# Entomopathogenic Nematodes and Their Symbiotic Bacteria from the National Parks of Thailand and Larvicidal Property of Symbiotic Bacteria against *Aedes aegypti* and *Culex quinquefasciatus*

**DOI:** 10.3390/biology11111658

**Published:** 2022-11-13

**Authors:** Aunchalee Thanwisai, Paramaporn Muangpat, Wipanee Meesil, Pichamon Janthu, Abdulhakam Dumidae, Chanakan Subkrasae, Jiranun Ardpairin, Sarunporn Tandhavanant, Timothy P. Yoshino, Apichat Vitta

**Affiliations:** 1Department of Microbiology and Parasitology, Faculty of Medical Science, Naresuan University, Phitsanulok 65000, Thailand; 2Centre of Excellence in Medical Biotechnology (CEMB), Faculty of Medical Science, Naresuan University, Phitsanulok 65000, Thailand; 3Center of Excellence for Biodiversity, Faculty of Sciences, Naresuan University, Phitsanulok 65000, Thailand; 4Department of Microbiology and Immunology, Faculty of Tropical Medicine, Mahidol University, Bangkok 10400, Thailand; 5Department of Pathobiological Sciences, School of Veterinary Medicine, University of Wisconsin, Madison, WI 53706, USA

**Keywords:** *Aedes*, bioassay, *Culex*, entomopathogenic nematode, *Photorhabdus*, *Xenorhabdus*

## Abstract

**Simple Summary:**

Entomopathogenic nematodes (EPNs) are insect-parasitizing nematodes of the genera *Heterorhabditis* and *Steinernema* that are symbiotically associated with the symbiotic bacteria *Photorhabdus* and *Xenorhabdus*, respectively. *Heterorhabditis indica*, *H. baujardi*, *Heterorhabditis* SGmg3, *Steinernema guangdongense*, *S. surkhetense*, *S. minutum*, and *S. longicaudum* were isolated from soil samples in the national parks of Thailand. For symbiotic bacterial isolates, *P. luminescens* subsp. *akhurstii*, *P. luminescens* subsp. *hainanensis*, *P. luminescens* subsp. *australis*, *Xenorhabdus stockiae*, *X. indica*, *X. griffiniae*, *X. japonica*, and *X. hominickii* were isolated from those EPNs. In mosquito larvicidal bioassays, *Photorhabdus* isolates were effective against both *Aedes aegypti* and *Culex quinquefasciatus*. In conclusion, a wide diversity of entomopathogenic nematodes and symbiotic bacteria was found in the national parks of Thailand. Moreover, isolated *Photorhabdus* bacteria were shown to have potential as biocontrol agents to control culicine mosquitoes.

**Abstract:**

Entomopathogenic nematodes (EPNs) are insect parasitic nematodes of the genera *Het-erorhabditis* and *Steinernema*. These nematodes are symbiotically associated with the bacteria, *Photorhabdus* and *Xenorhabdus,* respectively. National parks in Thailand are a potentially rich resource for recovering native EPNs and their symbiotic bacteria. The objectives of this study are to isolate and identify EPNs and their bacterial flora from soil samples in four national parks in Thailand and to evaluate their efficacy for controlling mosquito larvae. Using a baiting method with a *Galleria mellonella* moth larvae and a White trap technique, 80 out of 840 soil samples (9.5%) from 168 field sites were positive for EPNs. Sequencing of an internal transcribed spacer resulted in the molecular identification of *Heterorhabditis* nematode isolates as *H. indica*, *H. baujardi* and *Heterorhabditis* SGmg3, while using 28S rDNA sequencing, *Steinernema* nematode species were identified as *S. guang-dongense*, *S. surkhetense*, *S. minutum*, *S. longicaudum* and one closely related to *S. yirgalemense*. For the symbiotic bacterial isolates, based on *recA* sequencing, the *Photorhabdus* spp. were identified as *P. luminescens* subsp. *akhurstii*, *P. luminescens* subsp. *hainanensis* and *P. luminescens* subsp. *australis*. *Xenorhabdus* isolates were identified as *X. stockiae*, *X. indica*, *X. griffiniae*, *X. japonica* and *X. hominickii*. Results of bioassays demonstrate that *Photorhabdus* isolates were effective on both *Aedes aegypti* and *Culex quinquefasciatus*. Therefore, we conclude that soil from Thailand’s national parks contain a high diversity of entomopathogenic nematodes and their symbiotic bacteria. *Photorhabdus* bacteria are larvicidal against culicine mosquitoes and may serve as effective biocontrol agents.

## 1. Introduction

Entomopathogenic nematodes (EPNs), in the genus *Steinernema* Travassos, 1927, *Heterorhabditis* Poinar, 1976 and *Neosteinernema* Nguyen and Smart, 1994, are soil-dwelling and insect-parasitizing nematodes [1,2,3]. *Steinernema*, in the family Steinernematidae, and *Heterorhabditis*, in the family Heterorhabditidae, are symbiotically associated with bacteria in genera *Xenorhabdus* and *Photothabdus*, respectively, while the symbiotic bacterial species associated with Neosteinernema in the family Steinernematidae is still unclear. The EPNs have been successfully used in the biological control of soil-dwelling insect larvae [4,5]. The infective juvenile (IJ) is the invasive, non-feeding stage of EPNs that parasitized and kills the insect host, usually within 48 hours aided by their bacterial partners [6]. Several groups of economically important insect pests can be effectively controlled by using EPNs including the larval stages of termites, mustard sawfly, cabbage leaf webber, fig moth, rice stem borers, carob moth and white grub [5,7]. Thus, promoting the use of EPNs in agriculture has clear implications for environmental sustainability and food safety [4]. Surveying and accurately identifying native EPNs and their symbiotic bacteria provide important baseline data for applying the EPNs as biocontrol agents in a local area. Native EPNs more easily adapt to local environmental and ecological conditions during the application process.

Molecular techniques based on sequencing of specific genes have been used to identify species of EPNs. For example, DNA sequence analysis of the 18S rDNA gene was used to distinguish between the genera *Steinernema* and *Heterorhabditis*. However, it was unable to distinguish between species due to low 18S rDNA sequence variation [8]. By contrast, the sequencing of the 28S rDNA gene and the highly variable internal transcribe spacer (ITS) region of *Steinernema* [9] and *Heterorhabditis* [10], respectively, were effective in differentiating members of these genera at the species level. For symbiotic bacteria, molecular identification of these bacterial species has been based on the sequencing of the several genetic markers including the *16S rDNA*, 50S ribosomal protein L2 (*rplB*) gene, recombinase A (*recA*), DNA gyrase beta subunit (*gyrB*), DNA polymerase III subunit beta (*dnaN*), and glutamyl-tRNA synthetase (*gltX*) [11,12,13]. Analysis of the DNA sequences of the housekeeping gene, *recA*, was sufficient to discriminate between *Photorhabdus* and *Xenorhabdus* species [13]. 

The previously studied locations for surveying EPNs and symbiotic bacteria in Thai-land included roadside verges, areas of fruit crops, field crops, rice fields, and the banks of rivers and ponds [14,15,16,17], and typically have yielded low prevalence (<10%) and species diversity (<10 species) of EPNs and symbiotic bacteria. Surveys of EPNs and symbiotic bacteria in the national parks have resulted in several new records in Thailand [18,19,20]. In addition, at least two new species of EPNs were isolated from national parks in Vietnam [21,22]. The national parks in Thailand are defined as areas that contain vast natural resources of ecological importance or unique beauty, or that possess flora and fauna of special importance. We hypothesize that these largely uninhabited areas may be exceptionally good natural resources containing a diversity of previously undiscovered EPNs and bacterial symbionts. In the present study, four national parks, including Kaeng Krachan National Park in Phetchaburi Province, Namtok Samlan National Park in Saraburi Province, Phu Phan National Park in Sakhon Nakhon Province, and Huai Nam Dang National Park in Chiang Mai Province, were selected as primary sampling locations to search for EPNs and their symbiotic bacteria. In addition, randomly selected symbiotic bacterial species were laboratory tested to evaluate their larvicidal activity against the mosquitos, *Aedes aegypti* and *Culex quinquefasciatus*. *Aedes aegypti* is an important vector for the dengue virus, while *Cx. quinquefasciatus* serves as the main vector for filarial worms. Chemical insecticides are the primary methods used to control these mosquitoes. However, their use is associated with known adverse effects on humans, animals and the environment [23]. Moreover, repeated applications of insecticides can result in the creation and establishment of chemical resistant mosquito strains [24]. Therefore, the aims of the present study are to identify the entomopathogenic nematodes and their bacterial symbionts from national parks in Thailand and to evaluate the larvicidal activity of the selected symbiotic bacteria against *Ae. aegypti* and *Cx. quinquefasciatus*.

## 2. Materials and Methods

### 2.1. Study Sites

Four national parks in Thailand were selected for study sites to collect EPNs and symbiotic bacteria: Kaeng Krachan National Park (Phetchaburi Province, western Thai-land), Namtok Samlan National Park (Saraburi Province, central Thailand), Phu Phan National Park (Sakhon Nakhon Province, northeastern Thailand) and Huai Nam Dang National Park (Chiang Mai Province, northern Thailand) (Figure 1). The protocol for soil sample collection in the national parks was approved and permitted by the Department of National Park, Wildlife and Plant Conservation, Thailand (Permit number 0907.4/5514).

### 2.2. Collection of Soil Samples

Soil samples were randomly collected from 4 national parks of Thailand. We collected 840 soil samples from 168 soil sites by hand shovel. At each site, 5 soil samples were taken from an area of approximately 10 m^2^ at a depth of 5–10 cm. Approximately 300–600 g of each soil sample was taken by hand shovel and transferred to a plastic bag. Soil parameters for each sample, including temperature, pH, and moisture, were recorded using a soil survey instrument. Site location and soil texture were also recorded. GPS Navigation was used to determine the longitude, latitude and altitude of each site.

### 2.3. Isolation of EPNs from Soil Samples

The infective juvenile (IJ) stage of the EPNs was isolated from soil samples using larval *Galleria mellonella* baiting, as described by Bedding and Akhurst [25]. The G. mellonella cadavers were collected and placed into a White trap that was maintained at room temperature (25–30 °C) to allow emergence of infective EPN juveniles [26]. All soil samples were rebaited using fresh insect larvae to maximize EPN recovery. Emergent nematodes were collected and re-exposed to insect larvae to confirm entomopathogenicity and increase EPN yields. The nematodes were kept in a culture flask containing distilled water at 13–15 °C prior to molecular identification.

### 2.4. Species Identification of EPNs

For preliminary identification of nematodes, the skin color of *G. mellonella* cadavers was observed to predict the cause of death by *Steinernema* (beige/ochre or black) or *Heterorhabditis* (red, burgundy) [27]. The genomic DNA was extracted from approximately 200–500 IJs by using the NucleoSpin Tissue Kit (Macherey-Nagel, Duren, Germany) according to the manufacturer’s protocol. To check the quality of genomic DNA, five µl of purified DNA were examined on 0.8% agarose gel electrophoresis running in 0.5X TBE buffer at 80 V. After completion, the gel was stained with 10 µg/ml ethidium bromide for 1 min and destained with dH_2_O for 30 min. The DNA band was visualized, compared to a 100 bp molecular size marker and photographed under UV illumination. The genomic DNA was stored at −20 °C until use. 

Polymerase chain reaction (PCR) was performed for molecular identification and sequence analysis based on different gene targets for *Steinernema* and *Heterorhabditis*. Partial sequence of 28S rDNA for *Steinernema* nematodes was amplified using primers; 539_F (5′-GGATTTCCTTAGTAACTGCGAGTG-3′) and 535_R (5′-TAGTCTTTCGCCCCTATAC-3′) to obtain an 870 bp amplicon [9]. The reaction was carried out in a 30 µl volume containing 15 µl of EconoTaq^®^ PLUS 2× Master mix (1×; Luci-gen Corporation, Middleton, WI, USA), 1.5 µl of 5 µM of each primer (0.25 µM), 9 µl of dH20, and 3 µl of the DNA template (20–200 ηg). Partial internal transcribed spacer (ITS) sequence for *Heterorhabditis* nematode was amplified using the primers: 18S_F (5′-TTGATTACGTCCCTGCC CTTT-3′), TW81_F (5′-GTTTCCGTAGGTGAACCTGC-3′), 16S_R (5′-TTTCAC TCGCCGTTACTAAGG-3′) and AB28_R (5′-ATATGCTTAAGTTCAGCGGGT-3′) to obtain amplicons that varied in size between strains at a range of 800-850 bp for TW81_F and AB28_R primers and 1000–1100 bp for 18S_F and 16S_R or AB28_R primers [10]. The PCR components (30 µl total volume) were also used for amplifying *Steinernema* nematodes except for primers. All PCR reactions were performed in a Biometra TOne Thermal Cycler (Analytik Jena AG, Jena, Germany) using a temperature profile as previously described in Thanwisai et al. [15]. The amplified products were separated by 1.2% agarose-gel electrophoresis and visually examined. The PCR products were purified using a NucleoSpin^®^ Gel and PCR Clean-Up Kit (Ma-cherey-Nagel, Düren, Germany) according to the manufacturer’s instructions before sequencing at Macrogen, Inc. South Korea (http://www.macrogen.com, accessed on 6 August 2022).

### 2.5. Species Identification of Symbiotic Bacteria

*Xenorhabdus* and *Photorhabdus* were isolated from the haemolymph of the *G. mellonella* infected with the IJ of EPNs according to Fukruksa et al. [28]. Colony morphology of *Xenorhabdus* and *Photorhabdus* was observed on NBTA agar plates and examined for size, color, edge morphology and surface texture [15].

Preparation of bacterial cell genomic DNA extraction was performed according to the methods described by Yooyangket et al. [19] using a Genomic DNA Mini Kit (blood/Cultured Cell) (Geneaid Biotech Ltd., New Taipei City, Taiwan). One microliter of genomic DNA was examined using 0.8% agarose gel electrophoresis. 

The PCR mixture (30 μl total volume) targeting the *recA* gene contained 3 μl of 10× buffer (1×), 4.2 μl of 25 mM MgCl2 (3.5 mM), 0.6 μl of 10 mM dNTPs (200 μM), 1.2 μl of 5 μM from each primer (0.8 μM), 0.3 μl of 2.5-unit Taq DNA polymerase (0.1 U/ml), 3 μl of genomic DNA solution (20–200 ηg) and 16.5 μl of sterile distilled water. The recA primer sequences were recA1_F (5′-GCTATTGATGAAAATAAACA-3′) and recA2_R were (5′-RATTTTRTCWCCRTTRTAGCT-3′) [13]. The PCR reaction was performed in a Biometra TOne Thermal Cycler (Analytik Jena AG, Jena, Germany). PCR parameters for *recA* gene of *Xenorhabdus* were an initial denature step of 94 °C for 5 min, followed by 30 cycles of denaturation of 94 °C for 1 min, annealing temperature of 50 °C for 1 min and extension of 72 °C for 2 min with a final extension of 72 °C for 7 min. Parameters for *Photorhabdus* were an initial denature step of 94 °C for 5 min, followed by 30 cycles of denaturation of 94 °C for 1 min, annealing temperature of 50 °C for 45 s and extension of 72 °C for 1.5 min with a final extension of 72 °C for 7 min. The PCR products of *recA* of both genera (890 bp) were examined on 1.2% agarose gel electrophoresis. Twenty-nine microliters of PCR products were purified using Gel/PCR DNA Fragments Extraction Kit (Geneaid Biotech Ltd., Taiwan), as previously described by Yooyangket et al. [19]. The sequencing of *recA* gene was performed by Macrogen Inc. Service, South Korea (http://www.macrogen.com, accessed on 6 August 2022).

### 2.6. Analysis of the ITS, 28S rDNA, and recA Sequences

Chromatogram sequence ambiguity resolution was visually checked using SeqManII software (DNASTAR Inc., Madison, WI, USA). Species identification was performed using a BLASTN search against all nucleotide sequences (excluding human and mouse genomes) currently available in GenBank (http://blast.ncbi.nlm.nih.gov/Blast.cgi, accessed on 16 August 2022), and the match with the highest similarity score was selected. All nucleotide sequences of the ITS and 28S rRNA genes of EPNs and recA sequences of symbiotic bacteria were downloaded and aligned with our sequences using Clustal-W [29], which was included in the MEGA software version 7.0. [30]. Maximum likelihood trees were reconstructed using Nearest-Neighbor-Interchange (NNI) and Tamura-Nei model using MEGA software version 7.0 [30]. Bootstrap analysis was carried out with 1000 datasets.

### 2.7. Bioassay for Larvicidal Property against Aedes aegypti and Culex quinquefasciatus

Mortality rates of *Aedes aegypti* and *Culex quinquefasciatus* larvae (3rd–4th instar) were observed in the laboratory to measure bacterial larvicidal activity. The batched eggs of *Ae. aegypti* and the larvae of *Cx. quinquefasciatus* were obtained from the Medical Entomology Division, National Institute of Health, Department of Medical Sciences, Ministry of Public Health of Thailand. They were transported to the Department of Microbiology and Parasitology, Faculty of Medical Science, Naresuan University. The larvae of *Cx. quinquefasciatus* were maintained in dechlorinated water for one day prior to testing. The eggs of *Ae. aegypti* were allowed to hatch, and the first instar larvae were similarly maintained in dechlorinated water. The mosquito larvae were fed with minced pet food. The late 3rd and 4th instar larvae of both mosquitoes were used in the bioassays. 

Six isolates of symbiotic bacteria (2 isolates of *Xenorhabdus*: *Xenorhabdus* bPP39.5_TH, and *Xenorhabdus* bHND30.5_TH and 4 isolates of *Photorhabdus*: *Photorhabdus* bKKC20.5_TH, *Photorhabdus* bKKC25.3_TH, *Photorhabdus* bPP3.5_TH, and *Photorhabdus* bPP7.1_TH) were selected to test for their mosquito larvicidal potential. A single colony of *Photorhabdus* or *Xenorhabdus* isolate was selected from an NBTA plate and sterilely transferred to a 15 ml tube containing 5 ml of 5YS broth medium composed of 5% yeast extract (*w*/*v*), 0.5% NaCl (*w*/*v*), 0.05% K_2_HPO_4_ (*w*/*v*), 0.05% NH_2_H_2_PO_4_ (*w*/*v*), 0.02% MgSO_4_.7H_2_O (*w*/*v*) [31]. In the control group, a single colony of *Escherichia coli* ATCC 25922 on TSA agar was subcultured in 5YS broth and then processed under the same condition used for *Photorhabdus* and *Xenorhabdus*. 

Bioassays of both mosquito species were performed according to a previous study by Yooyangket et al. [19]. Thirty larvae in 3 wells of a 24-well plate (10 larvae/well) for each mosquito was tested against each symbiotic bacterial (10^8^ cfu/ml) isolate. The assay for each mosquito species was carried out 3 times. After incubation of test plates for 96 h at room temperature, the number of dead larvae was assessed based on observing no movement after teasing with a fine sterile toothpick.

### 2.8. Statistical Analysis

The survival of mosquito larvae, when exposed to the symbiotic bacteria isolated from EPNs, was compared with the controls (*E. coli* and distilled water). Analysis of the Log-rank test for equality of survivor functions was performed using the STATA version 13.0. A *p*-value less than 0.05 was considered as statistically significant differences between the 2 groups.

## 3. Results

### 3.1. Recovery of EPNs

A total of 80 out of 840 soil samples (9.5%) from 168 soil sites were positive with EPNs. More *Heterorhabditis* isolates (n = 47) than *Steinernema* (n = 33) were found in loam-like textures. Most EPNs were recovered from Phu Phan National Parks in Sakhon Nakhon province, Northeast Thailand. Twelve isolates for *Steinernema* and 18 isolates for *Heterorhabditis* were identified to the species levels (Table 1). However, other EPNs were not identified at the species level due to contamination with fungi during collection or poor sequencing data. The soil parameters of pH, temperature and moisture showed similar ranges between samples with EPNs and without EPNs (Table 2).

### 3.2. Identification and Phylogeny of EPNs

Eighteen isolates of EPNs were molecularly identified based on 680 bp of the ITS region for *Heterorhabditis* (GenBank accession numbers ON710863-ON710880). Eleven *Heterorhabditis* isolates were identified as *H. indica* with a high similarity score (98–100%), and two others, identified as *H. baujardi*, also exhibiting high identity scores (99–100%). In addition, five isolates of *Heterorhabditis* were similar to *Heterorhabditis* SGmg3 (97–99%) (Appendix A). A maximum likelihood tree of the *Heterorhabditis* isolates showed three main groups; the first group contained eleven *Heterorhabditis* isolates in the present study and one isolate of *H. indica* (accession number KP970842), the second group contained two *Heterorhabditis* isolates together with a *H. baujardi* (accession number MF618321), and the remaining group contained five *Heterorhabditis* isolates together with a *Heterorhabditis* SGmg3 (accession number FJ751864) (Figure 2). 

For *Steinernema*, 15 isolates (GenBank accession numbers ON715452-ON715466) were molecularly identified based on a 633 bp of the 28S rDNA. Twelve isolates of *Steinernema* were identified as *S. guangdongense* (five isolates) with 97% identity, *S. surkhetense* (three isolates) with 100% identity, and *S. minutum* (four isolates) with 98–99% identity. One isolate of *Steinernema* was closely related to *S. minutum* (95% identity), and one isolate of *Steinernema* was closely related to *S. longicaudum* (96% identity). The remaining isolate of *Steinernema* was closely related to *S. yirgalemense* (93% identity) (Appendix A). The phylogenetic tree of the *Steinernema* isolates was divided into five main groups: group one contained one *Steinernema* isolate in the present study and one isolate of *S. longicaudum* (accession number GU395644); group two contained three *Steinernema* isolates in the present study together with *S. surkhetense* (accession number MF621004); group three consisted of five *Steinernema* isolates in this study and an isolate of *S. minutum* (accession number GU64715); group four contained only one *Steinernema* isolate in the present study together with *S. yirgalemense* (accession number AY748450); and last group contained five *Steinernema* isolates in the present study together with *S. guangdongense* (accession number AY170341) (Figure 3). *C. elegans* (accession number JN636101) was used as an outgroup.

### 3.3. Identification and Phylogeny of Symbiotic Bacteria

*Photorhabdus* and *Xenorhabdus* were preliminarily discriminated by colony morphology after culture on NBTA for 3–4 days in the dark at room temperature. *Photorhabdus* (41 isolates) were light or dark green colonies with smooth edges and convex or umbonated surfaces on NBTA, whereas 24 isolates of *Xenorhabdus* were dark blue colonies with rough edges and convex or umbonated surfaces on NBTA. All sixty-five isolates of symbiotic bacteria were molecularly identified based on 588 bp of a partial sequence of the *recA* gene. *Photorhabdus* isolates (n = 41; GenBank accession numbers ON751626-ON751666) were identified as *P. luminescens* subsp. *akhurstii* (31 isolates), *P. luminescens* subsp. *hainanensis* (nine isolates) and *P. luminescens* subsp. *australis* (one isolate) with high identity ranging from 97% to 100% (Appendix A). Phylogeny showed most *Photorhabdus* isolates (n = 40) in the present study fell into group one, which contained *P. luminescens* subsp. *akhurstii* (accession number FJ862005) and *P. luminescens* subsp. *hainanensis* (accession number FJ862004), and only one isolate closely related to *P. asymbiotica* subsp. *australis* (accession number FJ862018) (Figure 4). 

For twenty-four *Xenorhabdus* isolates (GenBank accession numbers ON751667-ON751690), nineteen were identified as *X. stockiae* (ten isolates) and *X. indica* (two isolates) with 97–99% identity, *X. griffiniae* (one isolate) with 99% identity, *X. japonica* (five isolates) with 97–98% identity, and *X. hominickii* (one isolate) with 100% identity. The remaining five Xenorhabdus isolates in the present study were closely related to *X. ehlersii* (96% identity) (Appendix A). Phylogenic analysis of the *Xenorhabdus* isolates in the present study revealed a wider distribution as indicated by six groups. Group one contained five *Xenorhabdus* isolates that were closely related to *X. ehlersii* (accession number FJ823398); group two contained only one *Xenorhabdus* isolate related to *X. griffiniae* (accession number FJ823399); group three had five *Xenorhabdus* isolates related to *X. japonica* (accession number FJ823400); group four contained two *Xenorhabdus* isolates related to one *Xenorhabdus* isolate in the present study and *X. hominickii* (accession number FJ823410); and finally, group five contained two isolates of *Xenorhabdus* in this study together with *X. indica* (accession number FJ823420). Group six, the largest number of 10 Xenorhabdus isolates in this study, grouped together with one *X. stockiae* sequence (accession number FJ823425) downloaded from GenBank (Figure 5). *E. coli* (accession number U00096) was used as an outgroup.

### 3.4. Whole Cell Suspension of Symbiotic Bacteria against the Larvae of Aedes aegypti and Culex quinquefasciatus

Whole cell suspensions of symbiotic bacteria were tested against larvae of two mosquitoes, *Aedes aegypti* (Figure 6A) and *Culex quinquefasciatus* (Figure 6B), to evaluate their larvicidal activities. The larvae of *Ae. aegypti* began to die at 24 h postexposure to the symbiotic bacteria. The cumulative mortality of *Ae. aegypti* larvae was as high as 48.89% after exposure to *Photorhabdus* bPP7.1_TH (Appendix A). In contrast, the mortality in the control of *Ae. aegypti* larvae was at its lowest at 0% and 2.22% after exposure to *Escherichia coli* ATCC 25922 and distilled water for 96 h, respectively. All isolates of symbiotic bacteria tested against *Ae. aegypti* larvae showed significant differences compared to controls at *p*-value ≤ 0.05 (Appendix A). The larvae of *Cx. quinquefasciatus* exhibited a high mortality after exposure to *Photorhabdus* bPP3.5_TH for 24 h. The mortality of *Cx. quinquefasciatus* was similarly high (83.33–96.67%) after exposure to *Photorhabdus* isolates (bKKC25.3_TH, bPP3.5_TH and bPP7.1_TH) for 96 h. However, in the control, the mortality of *Cx. quinquefasciatus* was also high after exposure to *Escherichia coli* ATCC 25922 and distilled water for 96 h. Other isolates of symbiotic bacteria tested against *Cx. quinquefasciatus* larvae showed significant differences compared to control groups at *p*-value ≤ 0.05 (Appendix A).

## 4. Discussion

Several surveys of the local EPNs over a wide diversity of ecological locations have been reported worldwide, including Thailand [32]. However, there have been few studies specifically surveying EPNs in national parks. In 2005, *Steinernema robustispiculum*, a new species of EPN, was isolated from the woodlands of Chumomray National Park, Sason, Sathay, Kontum, Vietnam [21]. In 2014, *Steinernema huense*, a novel EPN species, was isolated from the soil in the forest in Bach Ma National Park, Thua Thien Hue Province, Vietnam [22]. In Thailand, several species of EPNs and symbiotic bacteria were isolated from 6 national parks: Mae Wong, Nam Nao, Thung Salaeng Luang, Phu Hin Rong Kla, Namtok Chat Trakan and Kaeng Chet Khwae National Parks. In the first survey of the EPNs in a Thai national park (Mae Wong National Park, Kamphaeng Phet Province), *Heterorhabditis indica*, *H. baujardi*, *H. zealandica*, *S. websteri*, and *S. kushidai* were identified. This survey recorded *H. zealandica* and *S. kushidai* for the first time in the country [18]. Yooyangket et al. reported the finding of *H. baujardi* and *S. websteri* in Nam Nao National Park, Phetchabun Province [19], while, most recently, *S. longicuadum* was isolated from soil samples in Kaeng Chet Khwae National Park in Phitsanulok Province, Thailand [20]. In the present study, *H. indica*, *H. baujardi*, *Heterorhabditis* SGmg3, *S. guangdongense*, *S. surkhetense*, and *S. minutum* were isolated from soil samples in Kaeng Krachan National Park, Phetchaburi Province, Namtok Samlan National Park, Saraburi Province, Phu Phan National Park, Sakhon Nakhon Province and Huai Nam Dang National Park, Chiang Mai Province, Thailand. This suggests that the forest soil environments of the national parks contain a wide diversity of EPN species. This diversity is due, at least in part, to a large number of different insect host species that coexist in Thailand’s national parks. The prevalence of EPNs isolated from soil samples in national parks of Thailand are variable, ranging from 2.87% in Nam Nao National Park [19], 4.36% in Mae Wong National Park [18], 9.1% in four national parks of Phitsanulok Province [20] to 9.5% in the present study. The presence of EPNs in the surveyed areas is influenced by several factors. Although soil pH, moisture and temperature between soil samples do not appear to be correlated with EPN presence or absence in this study, abiotic factors such as soil chemistry and pH, temperature, texture and structure and moisture are important for occurrence/distribution/survival of the EPNs [33]. In addition, biotic factors, e.g., natural predators/pathogens, interspecific or intraspecific competition among EPNs, and cooperation among EPNs can influence the EPN abundance and species diversity [33].

At present, *Steinernema* (100 species) and *Heterorhabditis* (16 species) are found globally in diverse geographical regions [32,34]. In an earlier study of EPNs in Thailand, one survey reported nematode distribution by genus only [14], while other studies identified *Steinernema* species, including the novel *S. siamkayai* [35] and *S. minutum* in southern Thailand [36]. Subsequently, *H. indica* hosting *Photorhabdus luminescens* was reported from northeastern and southern Thailand [37]. This was followed by the identification of numerous species found in one study included *S. websteri*, *S. khoisanae*, *H. indica*, *H. bacteri-ophora* and *H. baujardi* [15], and subsequently, *S. websteri* reported from lower northern Thailand [16,17]. At present, 15 EPN species representing a variety of ecological soil types in different geographic regions of Thailand have been reported including, *S. siamkayai*, *S. surkhetense*, *S. websteri* (synonym *S. carpocapsae*), *S. scarabiae*, *S. kushidai*, *S. minutum*, *S. khoisanae*, *S. longicaudum*, *S. guangdongense*, *S. huense*, *S. sangi*, *H. indica* (synonym *H. gerrardi*), *H. baujardi* (synonym *H. somsookae*), *H. bacteiophora* and *H. zealandica* [15,17,18,19,20,28,35,36,38,39,40,41]. In addition, three *Steinernema* isolates (*Steinernema* sp. YNd80, *Steinernema* sp. YNc215, and *Steinernema* eKK26.2_TH) and one *Heterorhabditis* isolate (*Heterorhabditis* sp. SGmg3) were unclassified as to species. In addition, isolates of a *Steinernema* sp. closely related to *S. minutum*, and others closely related to *S. longicaudum* and *S. yirgalemense* were reported in the present study. Confirmation of the identities of these *Steinernema* isolates requires further follow up morphological and molecular characterizations. In the present study, only a partial region of the 28S rDNA was used to identify EPNs. Additional nucleotide regions are suggested to clarify the taxonomic status of these EPN isolates (Steinernema closely related to *S. minutum* or *S. longicaudum* or *S. yirgalemense*). Taken together, the number of species thus far identified indicates a rich and diverse EPN fauna in Thailand. 

At present, 24 species of bacteria *Xenorhabdus* [13,42] and five species of *Photorhabdus* [12,13,42] have been documented worldwide. In Thailand, *X. stockiae*, *X. miraniensis*, *P. luminescens* subsp. *hainanensis*, *P. luminescens* subsp. *akhurstii*, *P. luminescens* subsp. *laumondii* and *P. luminescens* subsp. *australis* were reported from 13 provinces [15]. Additional symbiotic bacterial species were reported as *X. vietnamensis*, *X. indica*, *X. ehlersii*, *X. japonica*, *X. griffiniae*, *X. eapokensis*, *P. luminescens* subsp. *namnaonensis* and *P. temperata* subsp. *temperata* [18,19,20,28,38,40,43]. Recently, *Photorhabdus australis* subsp. *thailandensis* was reported as novel subspecies from Thailand [44]. In the present study, *X. hominickii* associated with an unidentified EPN represents the first record in Thailand. This bacterium, previously found associated with *Steinernema karii* in Kenya [45] and with *S. monticolum* in Korea [46], exhibits an expanded geographic range and adds to the diversity of symbiotic bacteria in Thailand. 

Symbiotic bacteria are capable of producing bioactive compounds including antimicrobial, antiparasitic, insecticidal and other cytotoxic compounds. These bacteria have been widely used for biological control of many insect pests, including lepidopterans [47], coleopterans [48] and mosquitoes [49,50]. Several symbiotic bacteria strains have been used to test mosquitocidal activity against the larvae of Aedes species [20,38,49,51]. In addition, some EPN strains were reported to be pathogens for the larvae of mosquitoes [41,52,53,54]. This indicated that both symbiotic bacteria and EPNs may be employed to control insects living in water and might be useful in the development of biocontrol agents for the control of mosquitoes and other disease vectors. Our findings confirm that *Ae. aegypti* and *Cx. quinquefasciatus* larvae were susceptible to selected symbiotic bacteria. *Aedes aegypti* larvae were most susceptible to *Photorhabdus* bPP7.1_TH, whilst *Cx. quinquefasciatus* larvae were most susceptible to several *Photorhabdus* isolates. This indicates that *Photorhabdus* isolates are superior in killing *Ae. aegypti* and *Cx. quinquefasciatus* than *Xenorhabdus* isolates. Previous reports, however, have shown that *X. ehlersii* and *X. griffiniae* also serve as effective larvicide microbes due to the high mortality against larval *Ae. aegytpi* [20,28]. Orally ingested cell suspensions from *Photorhabdus* spp. [55], *Photorhabdus luminescens* and *X. nematophila* [50] exhibited high toxicity against culicine mosquitoes, while several bioactive compounds/proteins from *Photorhabdus*, such as anthraquinones [56] and PirAB protein [57], have been shown to be toxic against mosquitoes. Similarly, several secreted proteins and secondary metabolites from *Xenorhabdus* spp., including phenethylamides and indol derivatives [58,59], xenorhabdins and xenooxides [59], xenocoumacins [60], benzylideneacetone [61] and iodine [62] were effective in controlling culicine mosquitoes [63,64]. Novel formulations of symbiotic bacterial toxins also have been studied. For example, synthesized gold and silver nanoparticles (AuNPs and AgNPs) coated with supernatant of *Photorhabdus luminescens* strain KPR-8B showed high mortality on *Ae. aegypti*, *An. stephensi*, and *Cx. quinquefasciatus* [65], while combining *Xenorhabdus* or *Photorhabdus* with Cry4Ba toxin from *Bacillus thuringiensis* enhanced larvicidal activity against *Ae. aegypti* [66]. Therefore, it continues to be important to identify and test strains or isolates of symbiotic bacteria to develop alternative or new strategies to control mosquito vectors of human disease.

## 5. Conclusions

In summary, we identified several species of entomopathogenic nematodes and their symbiotic bacteria from the national parks of Thailand, including several yet unidentified species. The finding of *Xenorhabdus hominickii* represents a new record of the symbiotic bacteria in Thailand. The EPNs and their symbiotic bacteria identified in this study from national parks of Thailand represent a diverse population worthy of further research. Several of the *Photorhabdus* bacteria discovered in park soil samples show potential to control *Ae. aegypti* and *Cx. quinquefasciatus*. These bacterial symbionts may be used for controlling the larvae of culicine mosquitoes through the development of bacteria-derived larvicides.

## Figures and Tables

**Figure 1 biology-11-01658-f001:**
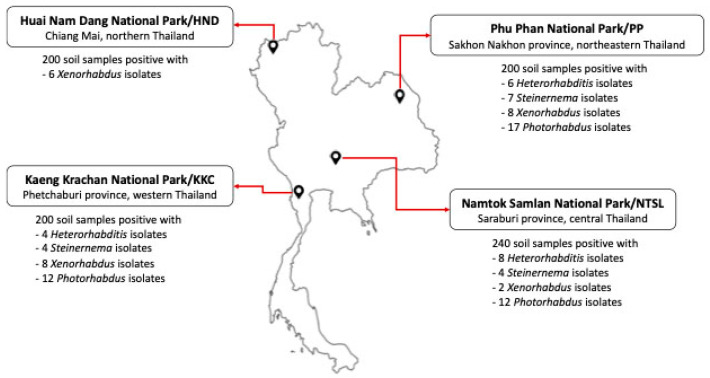
Map of Thailand shows the location of national parks where soil samples were collected. The number of identified positive EPN soil samples for each locality is listed.

**Figure 2 biology-11-01658-f002:**
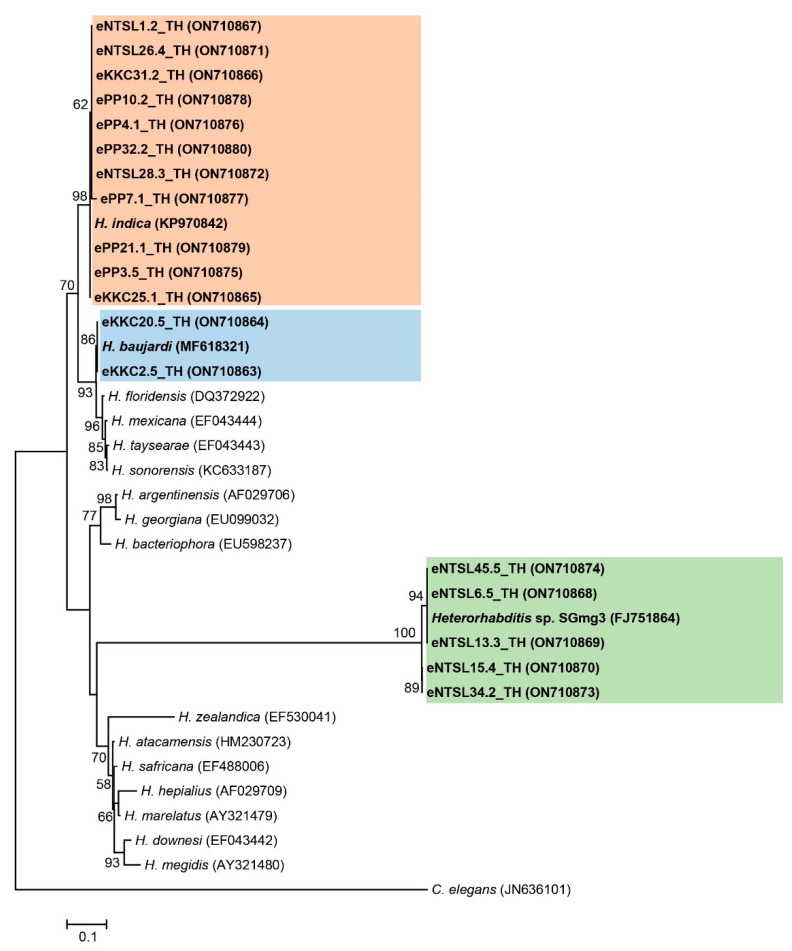
Maximum likelihood tree constructed based on a partial sequence of internal transcribe spacer (608 bp) of *Heterorhabditis* nematodes.

**Figure 3 biology-11-01658-f003:**
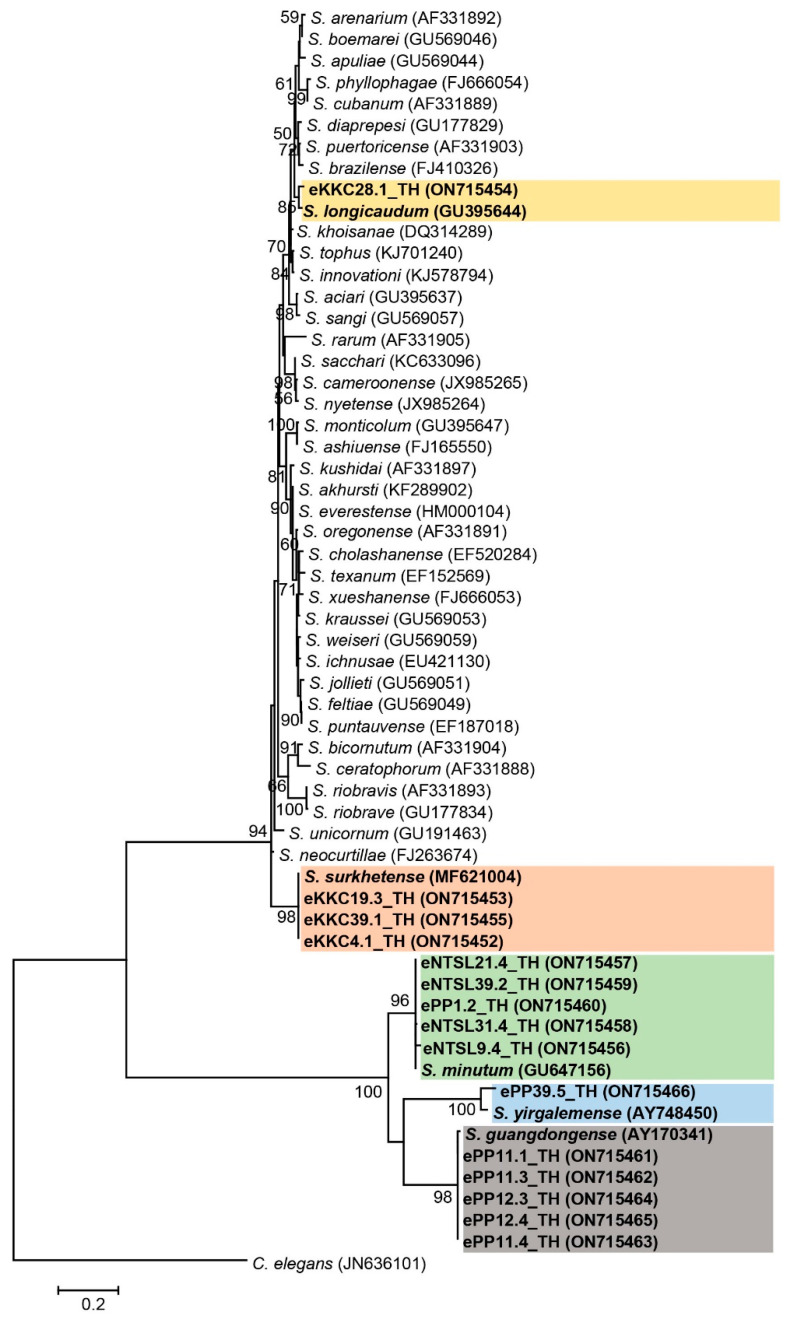
Maximum likelihood tree constructed based on a partial sequence of 28S rDNA (633 bp) of *Steinernema* nematodes.

**Figure 4 biology-11-01658-f004:**
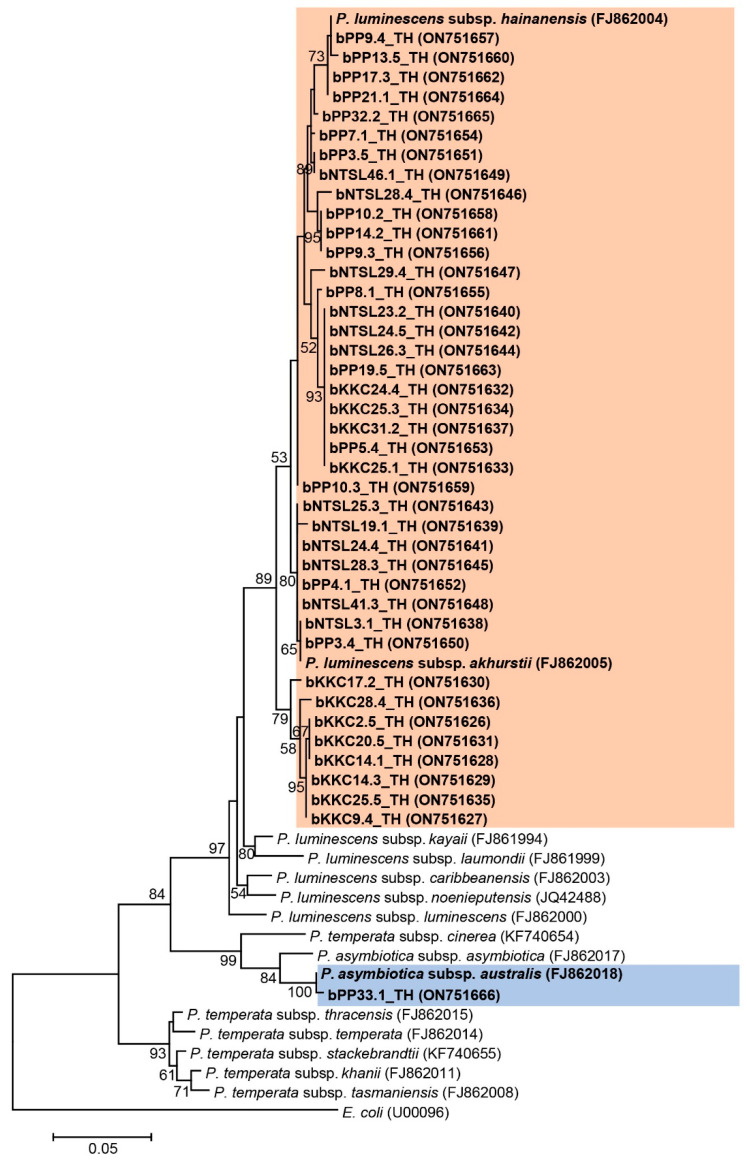
Maximum likelihood tree constructed based on a partial sequence of recA gene (588 bp) of Photorhabdus bacteria.

**Figure 5 biology-11-01658-f005:**
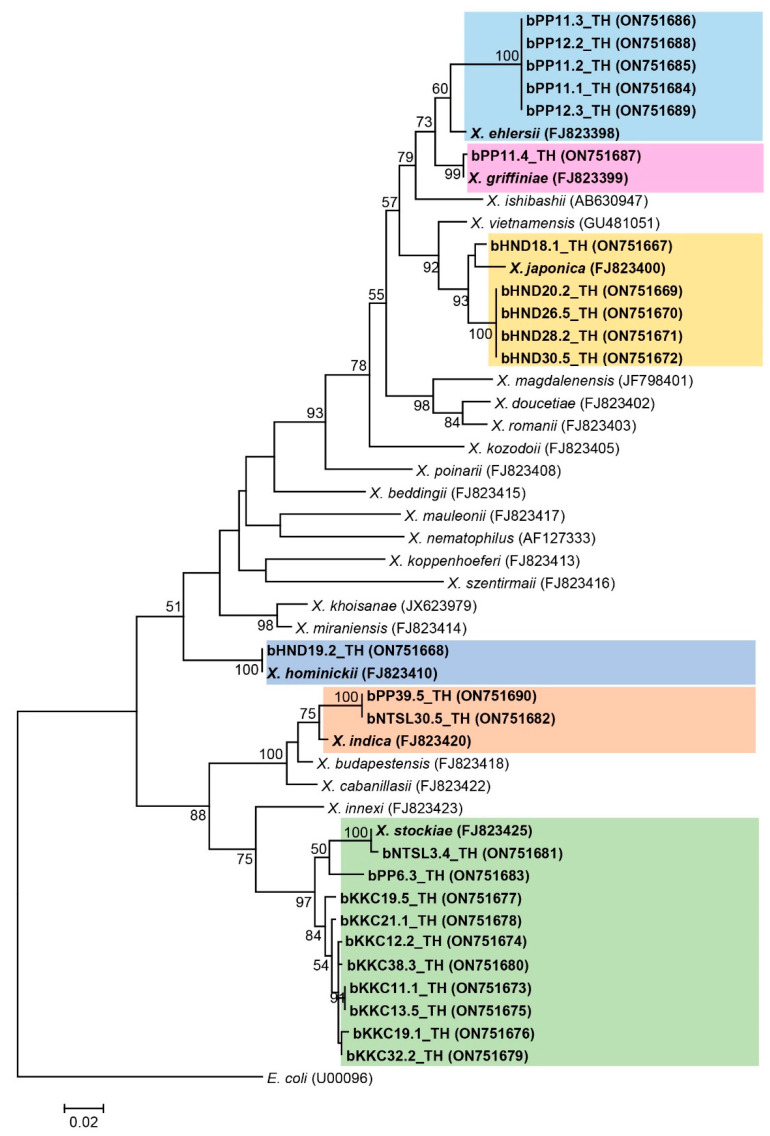
Maximum likelihood tree constructed based on a partial sequence of recA gene (588 bp) of Xenorhabdus bacteria.

**Figure 6 biology-11-01658-f006:**
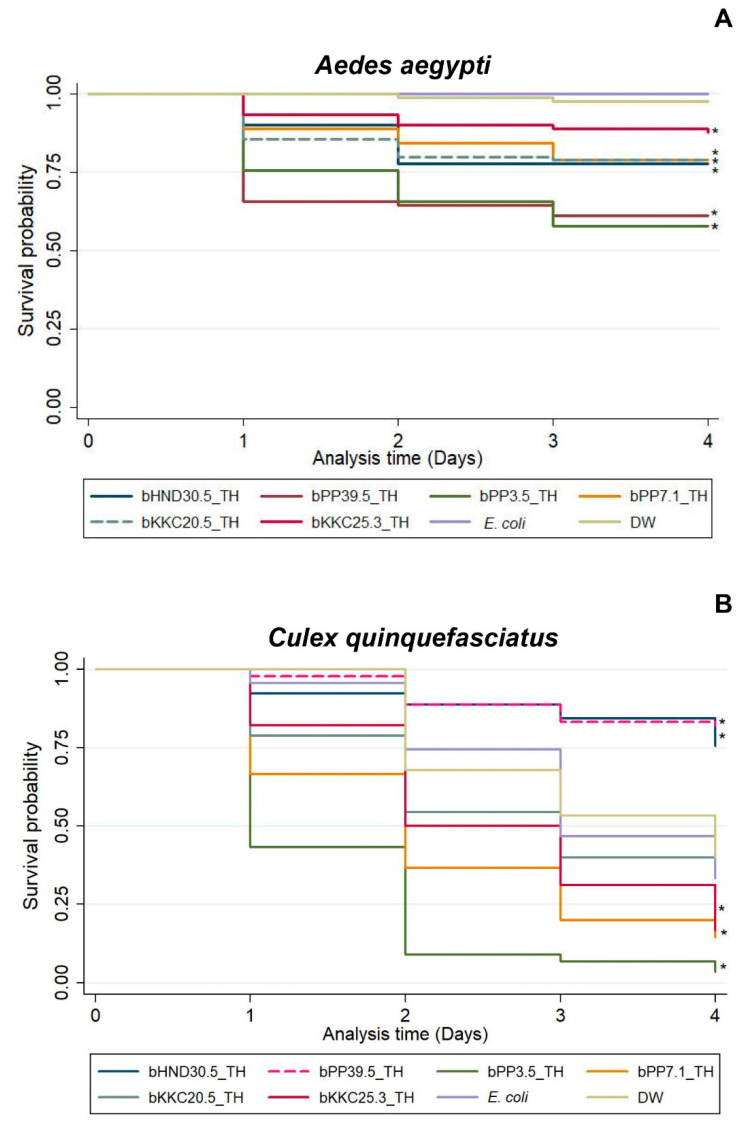
Survival analysis of the larvae of *Aedes aegypti* (**A**) and *Culex quinquefasciatus* (**B**) after exposure to the symbiotic bacteria isolated from entomopathogenic nematodes in national parks of Thailand. (bHND30.5_TH = *Xenorhabdus japonica*; bPP39.5_TH = *Xenorhabdus indica*; bPP3.5_TH = *Photorhabdus luminescens* subsp. *hainanensis*; bPP7.1_TH = *Photorhabdus luminescens* subsp. *akhurstii*, bKKC20.5_TH = *Photorhabdus luminescens* subsp. *akhurstii*; bKKC25.3_TH = *Photorhabdus luminescens* subsp. *akhurstii*; *E. coli* = *Escherichia coli* ATCC 25922; DW = Distilled water). Asterisks (*) indicate significant difference with control.

**Table 1 biology-11-01658-t001:** Number and identification of EPN/bacterial species recovery form soil samples in the national parks of Thailand.

National Park/Code	Province/Region of Thailand	No. of Soil Site	No. of Soil Samples (No. of Soil Sample with EPN)	No. of Positive with Molecular Sequences (Isolate)			
				Symbiotic Bacteria		EPNs	
				*Xenorhabdus*	*Photorhabdus*	*Steinernema*	*Hetorhabditis*
Huai Nam Dang National Park/HND	Chiang Mai/Northern	40	200 (6)	6	-	-	-
Kaeng Krachan National Park/KKC	Phetchaburi/ Western	40	200 (23)	8	12	4	4
Namtok Samlan National Park/NTSL	Saraburi/ Central	48	240 (24)	2	12	4	8
Phu Phan National Park/PP	Sakhon Nakhon/ Northeastern	40	200 (27)	8	17	7	6
Total		168	840 (80)	24	41	12	18

**Table 2 biology-11-01658-t002:** Soil pH, temperature and moisture of the samples collected from 4 of Thailand’s national parks.

National Parks/Code	Mean ± SD of Soil pH (Minimum–Maximum)			Mean ± SD of Soil Temperature (Minimum–Maximum)			Mean ± SD of Soil Moisture (Minimum–Maximum)		
	With EPN	Without EPN	All	With EPN	Without EPN	All	With EPN	Without EPN	All
Huai Nam Dang National Park/HND (n = 6 for with EPN, n = 194 for without EPN, n = 200 for all)	6.9 ± 0.11 (6.8–7.0)	6.87 ± 0.20 (5.8–7.0)	6.87 ± 0.19 (5.8–7.0)	21.17 ± 0.98 (20–23)	20.67 ± 1.08 (19–23)	20.68 ± 1.07 (19–23)	1.0 ± 0.00 (1.00)	1.12 ± 0.49 (1.0–5.0)	1.11 ± 0.48 (1.0–5.0)
Kaeng Krachan National Park/KKC (n = 23 for with EPN, n = 177 for without EPN, n = 200 for all)	6.75 ± 0.40 (5.6–7.0)	6.61 ± 0.58 (4.4–9.0)	6.62 ± 0.56 (4.4–9.0)	23.0 ± 1.00 (22–25)	22.79 ± 0.82 (22–26)	22.81 ± 0.83 (22–26)	1.87 ± 1.56 (1.0–7.0)	2.38 ± 2.04 (1.0–8.0)	2.31 ± 1.99 (1.0–8.0)
Namtok Samlan National Park/NTSL (n = 24 for with EPN, n = 216 for without EPN, n = 240 for all)	6.68 ± 0.21 (6.2–7.0)	6.62 ± 0.41 (4.2–7.0)	6.63 ± 0.39 (4.2–7.0)	26.21 ± 1.18 (24–28)	25.99 ± 1.03 (24–28)	26.0 ± 1.04 (24–28)	1.38 ± 0.58 (1.0–3.0)	1.40 ± 1.00 (1.0–8.0)	1.39 ± 0.96 (1.0–8.0)
Phu Phan National Park/PP (n = 27 for with EPN, n = 173 for without EPN, n = 200 for all)	6.61± 0.47 (5.0–7.0)	6.78 ± 0.38 (4.0–8.0)	6.75 ± 0.39 (4.0–7.0)	26.07 ± 0.87 (25–29)	26.43 ± 1.14 (20–30)	26.38 ± 1.11 (20–30)	1.98 ± 1.92 (1.0–7.0)	1.50 ± 1.33 (1.0–8.0)	1.56 ± 1.42 (1.0–8.0)

## Data Availability

All relevant data in this study are included in this published article and its supplementary information files. The nucleotide sequences that support the findings of this study have been deposited in GenBank under accession numbers ON710863-ON710880, ON715452-ON715466, ON751626-ON751666, ON751667-ON751690.

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
