# Peer review of "Entomopathogenic Nematodes and Their Symbiotic Bacteria from the National Parks of Thailand and Larvicidal Property of Symbiotic Bacteria against Aedes aegypti and Culex quinquefasciatus"

_biology, 2022, doi:10.3390/biology11111658_

Round 1

Reviewer 1 Report

The authors studied in the manuscript the isolation of EPNs and their symbiotic bacteria from some National Parks of Thailand and the larvicidal property of some symbiotic bacteria against two mosquito larvae.

As general considerations, I think that this study does not have the originality/novelty needed to be published in this journal. As the authors said in their manuscript, there are different papers about the isolation of EPNs from National Parks in Thailand and bioassay for larvicidal property of their bacteria against mosquito larvae:

Yooyangket, T.; Muangpat, P.; Polseela, R.; Tandhavanant, S.; Thanwisai, A.; Vitta, A. Identification of entomopathogenic nematodes and symbiotic bacteria from Nam Nao National Park in Thailand and larvicidal activity of symbiotic bacteria against Aedes aegypti and Aedes albopictus. PLoS One. 2018, 13, e0195681. https://doi.org/10.1371/journal.pone.0195681.

Thanwisai, A.; Muangpat, P.; Dumidae, A.; Subkrasae, C.; Ardpairin, J.; Tandhavanant, S.; Vitta, A. Identification of entomopathogenic nematodes and their symbiotic bacteria in national parks of Thailand, and mosquitocidal activity of Xenorhabdus griffiniae against Aedes aegypti larvae. Nematology. 2021, 24, 193-203.

Also, I think that one more time, related the isolation of EPNs and the mosquitocidal activity of EPN bacteria are objectives too different. The authors said in the discussion:  it continues to be important to identify and test strains or isolates of symbiotic bacteria to develop alternative or new strategies to control mosquito vectors of human disease”, in this case, why only 10 of the 65 isolates of these bacteria were tested against these mosquito species? This complete study could be the objective of a different manuscript.

I suggest that the present manuscript could be rewrite in two different papers, a short communication paper with the new isolates obtained in this complementary study of National Parks of Thailand, and a second paper with a more complete bioassays studding the property of all the bacteria isolates against the mosquito larvae.

Other suggestions:

The authors said that five soil samples were taken from an area of 10m2 and this is a soil sites.  In results they show the number of soil samples with nematodes, but we cannot see the number of soil sites with nematodes. If in a site sample all the five soil samples were positive of nematodes, all the nematodes are considered as five different isolates? Do you think that could be considered as different isolates all the nematodes found in a small area of 10m2? Could be considerer as the same population of nematodes and not as different isolates?

Finally review some mistakes in the abstracts and in other sections.

For exemple: “Entomopathogenic nematodes (EPNs), insect parasitic nematodes of genus Heterorhabditis and Xenorhabdus, are symbiotically associated with the bacteria, Photorhabdus and Xenorhabdus, respectively”. Change the first Xenorhabdus by Steinernema.

Author Response

Dear Reviewer 1 

Thank reviewer 1 for valuable suggestions and giving us a good chance to revise our manuscript. Our responses to your comments are indicated in red in the attached file

Best regards

Reviewer 2 Report

The paper is clear, easy to read, well designed and show a long laboratory and field work with good results. Just two comments: in materials and methods (line 166-170) please explain why DNA was checked before PCR. Delete comment in table 1.

Author Response

Dear Reviewer 2 

Thank reviewer 2 for valuable suggestions and giving us a good chance to revise our manuscript. Our responses to your comments are listed as the following;

- We thank reviewer for positive comments with our manuscript.

- We checked the genomic DNA by running on the 0.8 agarose gel electrophoresis because we would like to check the quality of the genomic DNA. We have rewrite the sentence (line 166-170) as “To check the quality of genomic DNA, five µl of purified DNA were examined on 0.8% agarose gel electrophoresis running in 0.5X TBE buffer at 80V.”

 - We have deleted comment in table 1 as reviewer suggested.

Best Regards 

Apichat Vitta

Reviewer 3 Report

The study is interesting and the results hold good importance in the control of two of the notorious disease causing mosquitos, A. aegypti and C. quinquefasciatus. The experiment hypothesis, methodology, and findings are well-executed and the data is presented in a good manner. I have a few suggestions for the authors to further improve the manuscript.

1.  Please revise the first sentences (line 32-34) of the abstract. Abstract should be the most clear and concise part of the manuscript. So, please avoid making long and complex sentences. Rather, make small and simple sentences conveying a clear message. 

2.  Line 48-50, just have a one liner conclusion depicting the most significant finding of the study. 

3. In introduction, after the first paragraph, please include information on the role of different sequencing approaches (28S rDNA, recA, etc.) in identifying the EPNs or the symbiotic bacteria.

4.  Merge paragraph 2 (line 100) and 3 (line 106) in the introduction as they talk about similar things.

5. In materials and method, have a separate sub-heading for statistical analysis and clearly mention the different statistical analysis (ANOVA and others) used in the study. 

6. For figure 2 and 3, I think the authors have used C. elegans as an outgroup. If yes, please mention the same in results section. 

7. Similarly, for figure 4 and 5, I think the authors have used E. coli as an outgroup. If yes, please mention the same in results section. 

8. For figure 6, please add a Y-axis information. Also, the authors have included replicates for the survival assays. So, the statistical significance should be shown in the graph. 

Author Response

Dear Reviewer 3 

Thank reviewer 3 for valuable suggestions and giving us a good chance to revise our manuscript. Our responses to your comments are  indicated in red in the attached file.

Best Regards 

Apichat Vitta 
